# Pilot Investigation on Markers of Bone Metabolism, Angiogenesis, and Neuroendocrine Activity as Potential Predictors of Survival of Metastatic Prostate Cancer Patients with Bone Metastases

**DOI:** 10.3390/ijms26104669

**Published:** 2025-05-13

**Authors:** Maria Angels Ortiz, Georgia Anguera, Elisabet Cantó, Jose Alejandre, Josefina Mora, Ruben Osuna-Gómez, Maria Mulet, Pradip Mora, Assumpta Antonijuan, Sofia Sánchez, Ona Ramírez, Vanessa Orantes, Pablo Maroto, Silvia Vidal

**Affiliations:** 1Inflammatory Diseases, Institut de Recerca Sant Pau (IR SANT PAU), 08041 Barcelona, Spain; mortiz@santpau.cat (M.A.O.); ecanto@santpau.cat (E.C.); jose.alejandre1996@gmail.com (J.A.); rosuna@santpau.cat (R.O.-G.); mmulet@santpau.cat (M.M.); 2Department of Medical Oncology, Hospital de la Santa Creu i Sant Pau, Universitat Autonoma de Barcelona, 08025 Barcelona, Spain; ganguera@santpau.cat (G.A.); ssanchez@santpau.cat (S.S.); oramirez@santpau.cat (O.R.); pmaroto@santpau.cat (P.M.); 3Department of Biochemistry, Hospital de la Santa Creu i Sant Pau, Universitat Autonoma de Barcelona, 08025 Barcelona, Spain; jmora@santpau.cat (J.M.); pmora@santpau.cat (P.M.); aantonijuan@santpau.cat (A.A.); vorantes@santpau.cat (V.O.)

**Keywords:** biomarkers, prostate cancer, bone metastasis

## Abstract

Prostate cancer with bone metastasis exhibits significant heterogeneity, complicating prognosis, and treatment. This study explores the potential of plasma, serum, and urine biomarkers to stratify patients and evaluate their prognostic value. Using two-step clustering, we analyzed baseline levels of Platelet-derived growth factor-BB (PDGF-BB), Insulin-like growth factor-binding protein 1 (IGFBP-1), Bone Morphogenetic Protein 2 (BMP-2), Vascular endothelial growth factor (VEGF) (plasma and urine), prostate-specific antigen (PSA), neuron-specific enolase (NSE), chromogranin A (CgA) and bone-specific alkaline phosphatase (BAP) (serum) and creatinine (Cr), and type I collagen-cross-linked N telopeptide (NTx) (urine) in 29 patients with prostate cancer and bone metastasis. Longitudinal biomarker dynamics were assessed at baseline, 6 months, and 12 months. Clinical outcomes were evaluated using Kaplan–Meier and multivariate analyses. Three distinct groups (C1, C2, and C3) were identified. C1 exhibited elevated pPDGF-BB and pVEGF levels, C3 had increased pBAP and uNTx/Cr, and C2 showed lower biomarker levels. Prior treatments influenced biomarker levels, with bisphosphonates reducing bone turnover markers and radiotherapy correlating with long-term changes in growth factors. Longitudinal analysis revealed unique biomarker dynamics within each group, with a tendency for pPDGF-BB and pVEGF levels to decrease over time in C1, and distinct trends in uNTx/Cr between groups. Despite individual biomarkers failing to predict survival, C3 patients demonstrated significantly worse survival than C1 and C2. Molecular clustering of peripheral blood and urinary biomarkers identifies distinct subgroups with metastatic castration-resistant prostate cancer patients outperforming traditional models in outcome prediction and supporting its potential for personalized treatment and prognosis.

## 1. Introduction

Prostate cancer frequently metastasizes to bone, profoundly impacting patient prognosis and quality of life [1]. Bone metastases are associated with increased skeletal-related events, reduced overall survival, and significant morbidity [2]. Current treatments for patients with metastatic castration-resistant prostate cancer (mCRPC) include active surveillance, surgery, radiotherapy, Radium-223 therapy, hormone therapy, bisphosphonates, and some investigational approaches [3]. Despite advances in treatment, substantial heterogeneity exists in disease progression and therapeutic response among patients [3]. This underscores the critical need to identify subgroups of patients with distinct molecular profiles and clinical outcomes to refine prognostic tools and tailor therapeutic approaches.

Patients with prostate cancer and bone metastasis exhibit elevated levels of various biomarkers related to bone metabolism and growth factors, many of which have demonstrated prognostic potential [4,5]. Bone formation markers, such as osteopontin (OPN), pro-collagen type I C-terminal and N-terminal pro-peptides (PICP and PINP), osteoprotegerin (OPG), and bone-specific alkaline phosphatase (BAP), are frequently elevated in these patients [4,6,7,8,9]. Bone resorption markers, including Type I collagen-cross-linked C-telopeptide (CTx), Type I collagen-cross-linked N-telopeptide (NTx), and bone sialoprotein (BSP), also correlate with disease activity [10]. Other markers, such as prostate-specific antigen (PSA), neuroendocrine markers like chromogranin A (CgA) and neuron-specific enolase (NSE), further reflect the complex pathophysiology of the disease [11,12,13]. Importantly, specific markers have shown clinical utility: BAP levels below 146 U/L are associated with improved survival in castration-resistant prostate cancer [14], and elevated PINP levels can predict bone metastases months before imaging detection [15]. Elevated circulating tumor cells (CTCs) and bone biomarkers are also linked to worse outcomes in patients receiving androgen deprivation therapy (ADT) [5,16]. Despite all this research, the analysis and interpretation of other molecules that reflect vascular remodeling and bone niche modification—such as VEGF, PDGF-BB, BMP-2, and IGFBP-1—as biomarkers remain complicated due to the influence of demographic characteristics, prior treatments, and disease-related factors. For example, treatments such as bisphosphonates or ADT can significantly alter biomarker levels, while variables like age, duration of therapy, and the extent of metastatic disease further contribute to variability [17,18]. These confounding factors must be carefully considered when using biomarkers for prognosis or subgroup classification, as they may obscure underlying disease-specific patterns.

While individual biomarkers provide valuable insights, a more integrative approach that combines multiple markers could better account for this complexity and enable the stratification of patients into distinct subgroups. Such strategies have the potential to improve predictions of clinical outcomes and guide personalized treatment approaches, ultimately enhancing the management of mCRPC. We hypothesize that mCRPC patients can be stratified into distinct molecular subgroups based on the peripheral blood and urinary biomarkers levels associated with bone metabolism, growth factors, and neuroendocrine activity. These subgroups are hypothesized to reflect the underlying heterogeneity of the disease, influenced by demographic characteristics, prior treatments, and the extent of metastatic disease. To test this hypothesis, firstly, we will employ two-step clustering to stratify distinct patient subgroups based on the combined profiles of selected biomarkers with potential prognostic relevance. Secondly, we will investigate the impact of clinical factors, such as demographic variables, disease activity, and prior treatments, on group formation and biomarker expression. Thirdly, we will analyze the longitudinal kinetics of these biomarkers within each group and evaluate whether group membership correlates with clinical outcomes, particularly overall survival. This integrative approach aims to inform biomarker-defined patient subgroups with distinct prognoses and support the development of personalized therapeutic strategies to improve patient outcomes.

## 2. Results

### 2.1. Clinical and Demographic Characteristics of Patients

The median age at diagnosis of the 29 included patients was 70.5 years (range: 58–85), and the median PSA level was 30.27 µg/L (range: 0.53–3645). Among the patients, 8 had a Gleason score < 8, 17 had a score ≥ 8, and this information was unavailable for 4 cases. At the time of diagnosis, 12 patients (41.38%) presented with metastatic disease. Regarding treatment, 11 patients (37.98%) had undergone radical prostatectomy, while 12 patients (41.38%) received local radiotherapy. The most common metastatic sites at the time of study inclusion were bone (24 patients), lymph nodes (16 patients), and lung (6 patients). Furthermore, 17 patients (58.6%) were treated with abiraterone acetate and prednisone.

We were unable to obtain plasma at t0 of one patient with 4 bone metastases and who had been treated with abiraterone + prednisone, ADT, chemotherapy, and bisphosphonates.

### 2.2. Identification of Molecular Clusters and Key Biomarkers

To explore the classification of 28 patients based on their biomarker profiles, we performed a two-step cluster analysis which automatically selected the most informative variables. The algorithm identified three patient groups based on four biomarkers selected from the initial panel of 14: pPDGF-BB, pVEGF, pBAP, and uNTx/Cr. The silhouette measure of cohesion and separation indicated ‘good’ cluster quality, suggesting meaningful differences between the groups. To better visualize the separation of patients into the three groups and highlight the distinct profiles, we have included a Principal Component Analysis (PCA) plot (Figure 1).

Levels of PDGF-BB, IGFBP-1, BMP-2, and VEGF were analyzed in peripheral blood and urine to evaluate their systemic concentrations and the quantities excreted through urine, aiming to test their potential as biomarkers. A comparison of peripheral blood and urinary levels revealed significant differences between the groups. Group 1 (C1) exhibited significantly higher levels of plasmatic PDGF-BB (pPDGF-BB) and VEGF (pVEGF) compared to group 2 (C2) and 3 (C3). In contrast, C3 had significantly higher levels of serum BAP (pBAP) than C1 and C2 and demonstrated elevated urinary NTx/Cr (uNTx/Cr) levels compared to C1. C2 appeared as a transitional group and consistently showed lower levels of these molecules, distinguishing it from the other two groups. Positive correlations were identified between pPDGF-BB and pVEGF, pIGF-BP1 and pCga, pVEGF and uCr, and pBAP and uNTx/Cr, suggesting a coordinated regulation or shared pathways influencing these biomarkers (Figure 2A). Conversely, pNSE levels exhibited negative correlations with pBAP and uNTx/Cr, indicating an inverse relationship between neuronal and bone turnover markers.

### 2.3. Influence of Clinical Factors and Prior Treatments on Biomarker Levels

To further investigate potential factors contributing to the molecular signatures observed in the groups, we compared clinical and demographic parameters (Table 1). No significant differences were found among the groups regarding patient age at diagnosis or at baseline, Gleason score at diagnosis, metastases at baseline or the frequency of treatments at baseline with abiraterone + prednisone, or bisphosphonates. Similarly, the time since diagnosis and the time from diagnosis to exitus or the frequency of different types of metastasis did not differ significantly between the groups. Examining the distribution of bone metastases (>4, <4, none) across groups (C1: 1, 5, 2; C2: 7, 3, 2; C3: 6, 1, 1) (Appendix A) reveals a trend: although not statistically significant, C3 predominantly includes patients with extensive bone disease, while C1 is mainly composed of those with limited or no bone involvement.

We next investigated factors that could influence the levels of the peripheral blood and urinary molecules analyzed (Table 2 and Appendix A). Although our initial aim was to analyze these factors within groups C1–C3, the analysis was conducted on the entire cohort due to insufficient sample sizes in some groups (C1 and C3) (Appendix A), which limited statistical interpretation.

Patients with prior treatment with bisphosphonates exhibited lower levels of pIGFBP-1, uIGFBP-1, and uNTx/Cr, suggesting a potential effect of bisphosphonates on these markers. Conversely, patients previously treated with abiraterone + prednisone had higher levels of uCr. Patients with previous surgery tended to have lower levels of pBAP and higher pNSE, but patients with radiotherapy or chemotherapy had lower levels of uPDGF-BB but higher uIGFBP-1, and higher levels of pNSE, respectively. Patients with more than four bone metastases exhibited higher levels of pBAP, suggesting increased bone turnover activity in this subgroup. In contrast, those with fewer than four bone metastases tended to have lower levels of pVEGF but higher levels of pBAP. Only a tendency to higher uVEGF was found in patients with higher Gleason index at the time of diagnosis.

### 2.4. Biomarker Dynamics

To assess the kinetics of soluble molecules in peripheral blood and urine, we analyzed available samples from patients at baseline (t0), 6 months (t1), and 12 months (t2) post-enrollment (Figure 3). The updated figure now indicates the timing and type of treatment changes for each patient. The small sample size in C3 (*n* = 2) in the longitudinal analysis limits the ability to generalize changes in biomarkers. As a result, statistical comparisons were restricted to C1 and C2. In patients of C1, we observed a tendency for pPDGF and pVEGF levels to decrease by t2 compared to baseline, and uNTx/Cr to increase by t2 compared to t1. No significant changes in pBAP levels were detected at either t1 or t2 compared to baseline. In patients of C2, no significant changes were detected in pPDGF, pVEGF, and pBAP levels at either time point relative to baseline, but a significant decrease in uNTx/Cr from t0 to t1 was found. The dynamics in the subgroup of patients who did not change their treatment between baseline and t2 were comparable to those who switched to enzalutamide or other treatments. Importantly, these biomarker dynamics were consistently observed within each group regardless of whether patients maintained or changed their therapy during follow-up. Patients who switched to enzalutamide or other treatments showed similar biomarker trends to those who remained on the same regimen. Biomarker uNTx/Cr trends were still distinct across groups C1 and C2, suggesting group-specific molecular profiles, rather than these treatment changes or their timing.

### 2.5. Prognostic Implications

To evaluate the prognostic potential of pPDGF, pVEGF, pBAP, and uNTx/Cr, we assessed survival outcomes based on their levels at baseline. Kaplan–Meier survival analysis showed no significant differences in survival when comparing patients in the highest quartile (4th quartile) of each marker to those in lower quartiles (Figure 4A). Similarly, comparing survival days across quartiles for each marker revealed no significant differences. In contrast, analysis by cluster demonstrated distinct survival outcomes (Figure 4B). C3 patients exhibited significantly shorter survival compared to C1 and C2. ANOVA confirmed a significant difference in survival among the three groups, specifically between C1 and C3. A multivariate analysis integrating the four markers (pPDGF, pVEGF, pBAP, and uNTx/Cr) failed to generate a significant overall model for predicting survival. Notably, within this model, pBAP emerged as the only significant individual marker. When comparing groups using a multivariate approach, the model was significant, and pairwise comparisons between groups also reached statistical significance. The area under the curve (AUC) analysis revealed that clustering outperformed the multivariate model based on the four markers in predicting survival. This underscores the prognostic value of patient stratification into groups, particularly in differentiating outcomes between C1 and C3. However, we recognize that the limited sample size of some groups, especially C1 and C3 (*n* = 8), may influence the robustness of the survival curves. These findings, while statistically significant, should be interpreted with caution and validated in larger, independent cohorts.

## 3. Discussion

This study highlights the molecular heterogeneity among mCRPC patients underscoring the potential of biomarker-based stratification to elucidate disease complexity, track progression, and improve prognostic predictions.

The molecular profiles of patient groups suggest differential regulation of pathways involved in bone metabolism, angiogenesis, and neuroendocrine activity. C1 was characterized by higher levels of pPDGF-BB and pVEGF, indicative of enhanced growth factor activity potentially reflecting an angiogenic and tumor-promoting microenvironment. In contrast, C3 exhibited elevated levels of pBAP and uNTx/Cr, markers of bone turnover, aligning with the observed tendency for higher prevalence of bone metastases at baseline. C2 displayed consistently lower biomarker levels, setting it apart as a potentially less active molecular state between the angiogenesis-dominant pattern of C1 and the bone turnover-dominant signature of C3. The absence of significant differences in demographic and clinical factors, including age, Gleason score, and prior treatment frequency, suggests that these groups represent inherent molecular characteristics rather than simple reflections of clinical variables. However, the trend toward higher bone metastatic burden in C3 supports the hypothesis that disease extent influences biomarker expression and survival [19].

The correlation between PDGF-BB and VEGF may reflect their synergistic role in neovascularization [20]. The correlations between pIGFBP-1 and pCgA, as well as between pVEGF and uCr, indicate mechanisms associated with poorer prognosis [21,22,23]. The negative correlation between pNSE and both pBAP and uNTx/Cr suggests reduced involvement of bone metabolism in tumors with a more neuroendocrine phenotype [24].

Our analysis revealed that prior treatments significantly influenced biomarker levels, underscoring the need to account for these factors in interpreting molecular data. Bisphosphonates were associated with reduced levels of pIGFBP-1, uIGFBP-1, and uNTx/Cr, consistent with their known effects on bone turnover. By suppressing osteoclast activity, bisphosphonates lower the release of IGF and its binding proteins from the bone matrix, leading to reduced levels in both plasma and urine [25,26]. Additionally, bisphosphonates that inhibit bone resorption decreased NTx/Cr levels in urine [27,28]. This is consistent with their role in stabilizing bone turnover and preventing skeletal-related events [29]. Conversely, radiotherapy was associated with increased uIGFBP-1 levels, possibly related to treatment-induced tissue remodeling [30,31]. Treatment with abiraterone + prednisone was correlated with higher uCr levels, likely reflecting its complex metabolic effects, including potential impacts on renal function despite corticosteroid presence [32]. In this line, Attard et al. reported an increased creatinine levels in some patients despite that abiraterone and prednisone treatment [33]. Chemotherapy may contribute to elevated NSE release, which could indicate neuroendocrine cell activity or stress [34]. Additionally, the elevated levels of pBAP and pVEGF in patients with metastatic disease suggest that these biomarkers may reflect tumor burden or skeletal involvement.

The timing of treatments also influenced biomarker dynamics. Time since radiotherapy was positively correlated with pIGFBP-1 and uNTx/Cr levels and negatively correlating with pVEGF. Radiotherapy can modulate tissue repair mechanisms and bone turnover, which may result in elevated uNTx/Cr levels over time due to compensatory osteoclast activity [35,36]. Conversely, time since androgen deprivation therapy was negatively associated with uVEGF, potentially reflecting its direct or indirect effects on angiogenesis and tumor suppression [37]. During the longitudinal analysis, biomarker changes were consistent within each group, but distinct across groups, indicating that underlying group characteristics drove the biomarker dynamics rather than treatment changes. These findings underscore the importance of considering treatment timing when interpreting the molecular profiles of mCRPC patients.

In our study, individual biomarker levels were not predictive of survival. However, clustering analysis revealed significant survival differences, with C3 exhibiting markedly worse outcomes than C1 and C2. This was further supported by multivariate analysis, where the clustering model outperformed models based on individual markers, as reflected by higher AUC values. These findings highlight the importance of integrating biomarker data to capture disease heterogeneity and improve prognostic accuracy. There are distinct molecular signatures for each group that may reflect differences in underlying biological processes between these patients. Patients in C3 with the worst outcome had the lowest PDGF-BB and VEGF levels and failed to support effective vascularization and tissue repair, potentially leading to poorer healing of bone lesions caused by metastases [38,39]. This insufficiency could also contribute to inadequate delivery of therapies to metastatic sites, accelerating disease progression. However, the small numbers of patients in the group may also lead to unstable estimates and exaggerated differences in Kaplan–Meier curves.

On the other hand, high pBAP and uNTx/Cr levels in patients of this group indicate heightened osteoblastic and osteoclastic activity, respectively, a state of aggressive bone turnover [40]. In the context of bone metastasis, this may reflect extensive bone destruction by metastases (osteolysis) and a compensatory but inefficient increase in bone formation. Such high turnover can weaken skeletal integrity, lead to more severe skeletal-related events, and release growth factors from the bone matrix that promote further tumor growth and metastasis. Urinary NTx/Cr has been linked to poor outcomes in patients with bone metastases, as it reflects high osteoclast activity and tumor-induced bone damage [41]. This metabolic state can result in systemic complications, such as hypercalcemia and bone pain, further reducing the quality of life and survival [42].

This study has several limitations. First, due to constraints in sample expansion, not all relevant biomarkers involved in bone metastasis and immune regulation were included, potentially limiting the molecular characterization. Future studies will address this by expanding the biomarker panel and validating findings in larger cohorts. Second, the small and uneven sample sizes across groups—particularly in C1 and C3—may affect the robustness and reproducibility of the classification. Larger, more balanced cohorts are needed to confirm the stability of the clustering. Additionally, the lack of pre-treatment samples limits the ability to assess baseline biomarker levels and may introduce confounding variables. Despite these limitations, the internal consistency of biomarker profiles and significant multivariate and pairwise results support the relevance of this stratification. As this is an exploratory study, further validation—including cross-validation and external cohorts—is needed to confirm the generalizability and prognostic value of these groups

Our results suggest a potential for grouping mCRPC patients based on selected molecular markers to predict outcomes; however, additional research is required to elucidate and validate this model in larger and independent cohorts. Clustering provides a more comprehensive understanding of disease heterogeneity, linking biomarker profiles to clinical outcomes and identifying patients of C3 at higher risk of poor survival. Incorporating biomarker dynamics and treatment history enhances this approach, paving the way for personalized therapeutic strategies. Future research should validate these findings in larger cohorts and explore the mechanistic basis of these molecular differences to identify actionable therapeutic targets.

## 4. Material and Methods

### 4.1. Patients

This prospective, non-interventional, exploratory study included patients diagnosed with metastatic castration-resistant prostate cancer (mCRPC). The inclusion period for the study started at baseline (t0). Patients were eligible for inclusion if they had a confirmed diagnosis of metastatic castration-resistant prostate cancer (mCRPC), an estimated life expectancy exceeding 12 weeks, an ECOG performance status of 1 or less, and no significant laboratory abnormalities in hemoglobin, lymphocyte count, platelet count, or creatinine levels. Exclusion criteria encompassed the presence of brain metastases and current enrollment in or eligibility for a clinical trial. All 29 consecutive patients who met the inclusion criteria of mCRPC were enrolled at baseline. The diagnosis was recorded retrospectively to assess the time since diagnosis and prior treatments. All patients were treated with either luteinizing hormone-releasing hormone (LHRH) agonists or underwent orchiectomy, ensuring testosterone levels were below 50 ng/dL. Patients included in the study were treated with first-line ARPIs, which is the standard of care. Radium-223 is typically used in later treatment lines; therefore, none of the patients had received Radium-223 prior to their inclusion in the study. The duration of the different previous treatments received by the patients at the time of inclusion (baseline, t0) is detailed in Appendix A. Urine and peripheral blood samples were collected from each patient, and plasma, serum, and urine were stored at −80 °C for subsequent analysis. The study was conducted at the Hospital de la Santa Creu i Sant Pau in Barcelona, Spain, between June 2020 and February 2022. Written informed consent was obtained from all participants, and the study protocol received ethical approval from the Institutional Ethics Committee of the Hospital de la Santa Creu I Sant Pau (IIBSP-P120-2018-20). Peripheral blood and urine samples from each patient were collected at the time of their inclusion in the study (t0). In a subgroup of patients, additional samples were collected 6 (t1) and 12 (t2) months after inclusion.

### 4.2. Determination of Plasma, Serum, and Urine Markers

Peripheral blood samples were collected in serum separator tubes and in plasma EDTA tubes and centrifuged at 3000× *g* for 15 min. PDGF-BB (Platelet-derived growth factor-BB), IGFBP-1 (Insulin-like growth factor-binding protein 1), BMP-2 (Bone Morphogenetic Protein 2), and VEGF (Vascular endothelial growth factor) were quantified in both plasma and urine using ELISA. All measurements were carried out using manufacturer-provided kits from Peprotech (Waltham, MA, USA). The measuring ranges for the assays were 15.62–1000 pg/mL for PDGF-BB, 23.43–1500 pg/mL for IGFBP-1, 46.87–3000 pg/mL for BMP2, and 62.5–4000 pg/mL for VEGF. Serum concentrations of PSA and NSE were measured using automated electrochemiluminescence immunoassays on the Roche Cobas^®^ e801 platform (Roche Diagnostics GmbH, Mannheim, Germany) with a measuring range of 0.006–100 μg/L for PSA and 0.075–300 μg/L for NSE. Detection limits were 0.010 μg/L for PSA and 0.15 μg/L for NSE. The serum concentrations of CgA were measured using automated immunofluorescent assay based on TRACE^®^ Technology (Time-Resolved Amplified Cryptate Emission) on the Kryptor^®^ platform (Brahms GmbH, Brandenburg, Germany), with a measuring range of 11.8–3000 μg/L and a detection limit of 11.8 μg/L. Serum concentrations of BAP were measured using a manual solid phase immunoenzymetric assay (Ostase^®^ BAP EIA, Immunodiagnostic System Ltd., Boldon, UK), with a measuring range of 0.1–94.5 μg/L and a detection limit of 0.7 μg/L. Urinary concentrations of the type I collagen-cross-linked N telopeptide (NTx) were measured using a manual solid phase immunoenzymetric assay (Osteomark^®^ NTx, Alere Inc, Scarborough, ME, USA), with a measuring range of 20–3000 nM BCE (bone collagen equivalents) per mM of creatinine and a detection limit of 1 nM BCE per mM de creatinine. Urinary concentrations of Creatinine were measured using an automated colorimetric Jaffe’s reaction with picric acid on the Alinity^®^ platform (Abbott, Longford, Ireland) with a measuring range of 0.2–74.2 mmol/L and a detection limit of 0.2 mmol/L. All molecules measured in peripheral blood (plasma or serum) are denoted with “p”, while those measured in urine are denoted with “u”.

### 4.3. Statistical Analysis

The Kolmogorov–Smirnov test was used to analyze data with a normal distribution. Qualitative variables were expressed as numbers and/or percentages, and quantitative variables were expressed as mean and standard deviation or median and range. Comparisons between groups were tested with the Student’s *t*-test or the Mann–Whitney test, according to a Gaussian distribution. Differences in categorical variables were assessed using the Chi-square test. Correlation analyses were carried out using Spearman correlations. R-Studio v. 4.0 was used to generate the correlograms. Kaplan–Meier survival analysis was used to evaluate the outcomes and the log-rank test was used to compare survival curves for different patient groups. Biomarker profiles were stratified into distinct patient groups using a multivariate two-step clustering algorithm. Two-Step Cluster Analysis was performed in SPSS v. 21 using a log-likelihood distance measure and automatic selection of the number of groups. An initial model including 14 plasma, serum, and urinary biomarkers from 28 patients yielded a silhouette score of 0.3, indicating poor-to-fair clustering quality. Predictor importance analysis identified the four most influential variables. To improve clustering performance and reduce dimensionality, a second analysis was performed using only these four variables. The clustering quality improved substantially, with a silhouette score of 0.6, indicating good cohesion and separation. Both 3- and 4-group solutions yielded the same silhouette value, but the 4-group solution included a small group (*n* < 5), which was considered statistically unreliable. Therefore, the 3-group model was selected for further analysis and visualization. All 14 candidate biomarkers were initially included, and the algorithm automatically selected the subset of variables that most contributed to distinguishing groups of patients. The optimal number of groups was determined based on the silhouette score, and clustering was performed accordingly on the selected variables. Statistical significance was defined as *p* < 0.05. The analyses were performed using Graph Pad Prism 9 and SPSS v.21 software.

## Figures and Tables

**Figure 1 ijms-26-04669-f001:**
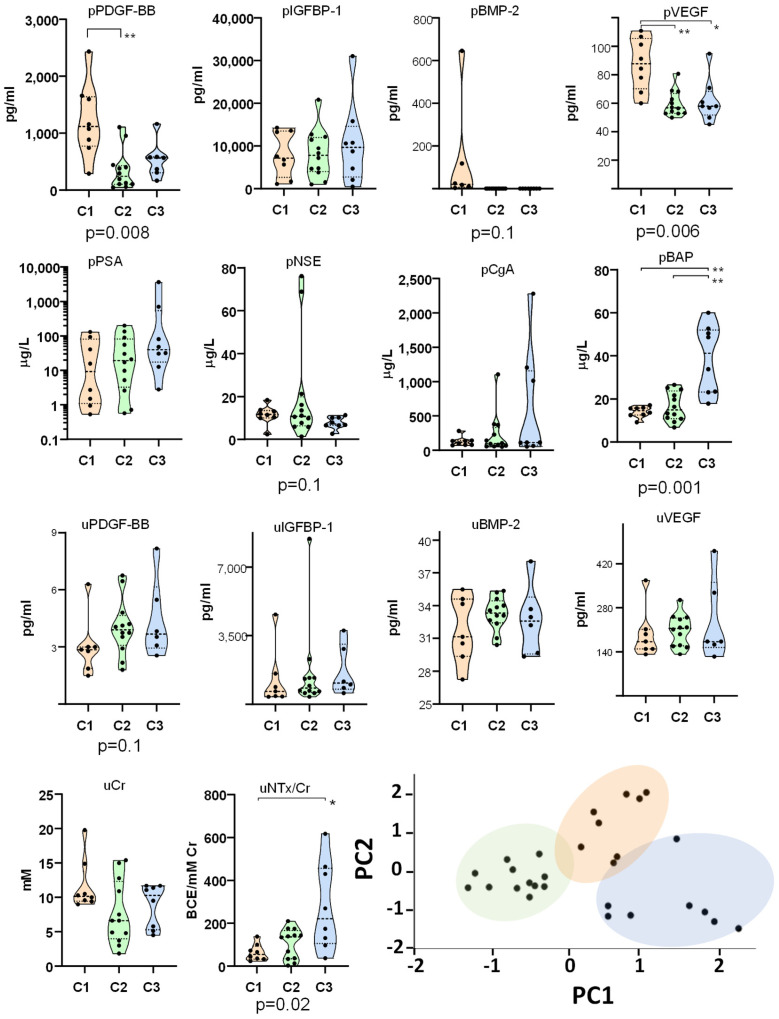
Peripheral blood (p) and urinary (u) levels of PDGF-BB, IGFBP-1, BMP-2, VEGF, PSA, NSE, CgA, BAP, Cr, and NTx/Cr. Biomarker levels were measured in 28 mCRPC patients stratified into three groups (C1, C2, and C3). A principal component analysis (PCA) plot is included to illustrate the distribution of patients across the identified groups. *p*-values for the comparison of groups using the Kruskal–Wallis test are shown below the graphs, with significant differences (*p* < 0.05) and trends (*p* = 0.1) highlighted. Pairwise differences are indicated on the graphs. (* *p* < 0.05, ** *p* < 0.01).

**Figure 2 ijms-26-04669-f002:**
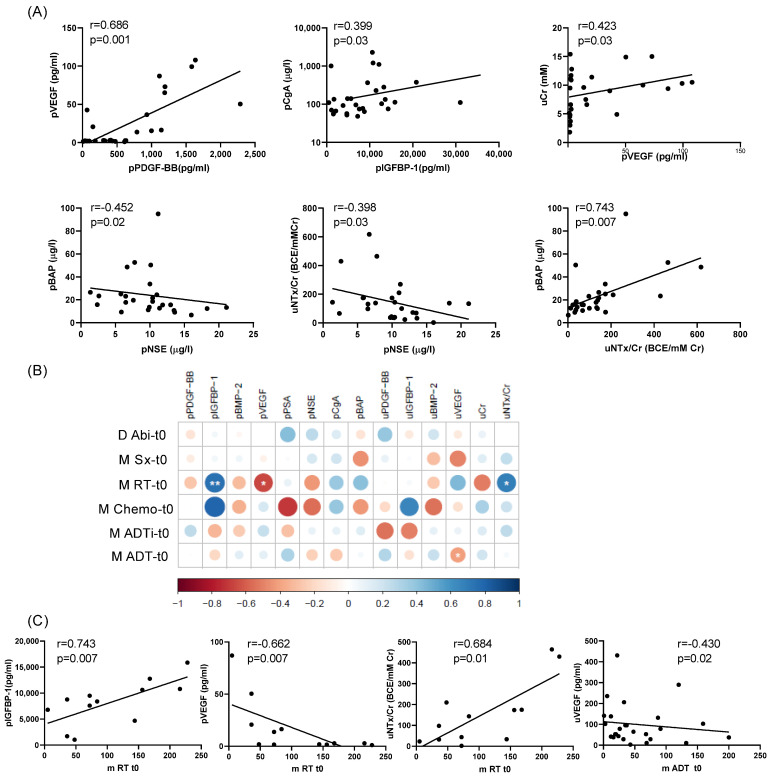
Correlations between peripheral blood and urinary levels of studied molecules and clinical variables at baseline. (**A**). Significant correlations among pPDGF-BB, pIGFBP-1, pVEGF, pNSE, pCgA, pBAP, uCr, and uNTX/Cr (peripheral blood levels denoted as “p” and urinary levels as “u”). (**B**). Correlogram showing relationships between molecules and clinical variables at baseline: days of Abiraterone + prednisone treatment (D Abi), months since surgery (M Sx), since radiotherapy (M RT), since chemotherapy initiation (M Chemo), of intermittent androgen deprivation therapy (M ADTi), and of continuous androgen deprivation therapy (M ADT). The correlation coefficients (r) were represented by circles of different sizes and color intensity indicating statistical strength. Blue circles are for positive correlations and red circles for negative correlations. Asterisks denote statistical significance (* *p* < 0.05, ** *p* < 0.01). (**C**). Graphical representation of significant correlations indicated in correlogram. Spearman’s test correlations were performed including all patients (*n* = 29) but pPDGF-BB, pIGFBP-1, pBMP-2, and pVEGF, data were from 28 patients. Significance was defined as *p* < 0.05.

**Figure 3 ijms-26-04669-f003:**
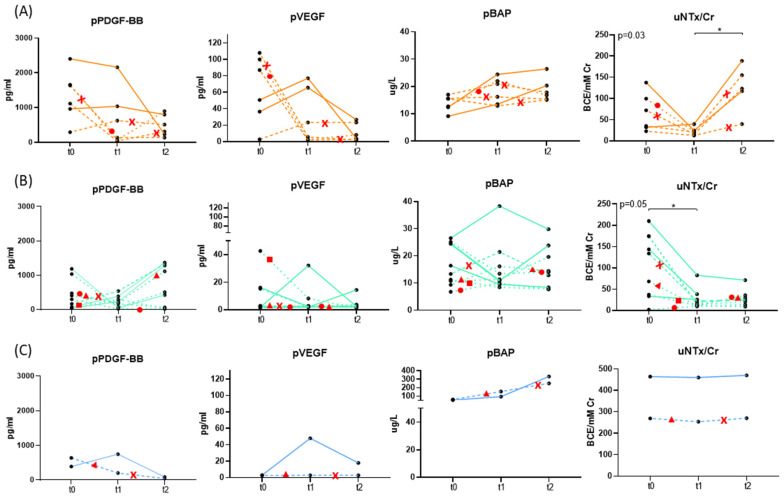
Peripheral blood levels of PDGF-BB, VEGF, BAP, and urinary levels of NTX/Cr over time (t0, t1, t2) in mCRPC patients segregated into groups C1 (**A**), C2 (**B**), and C3 (**C**). In all graphs, solid lines represent patients who did not change their treatment, dashed lines represent patients who switched to Enzalutamida, and dotted lines represent patients who switched to another treatment (non Enzalutamida) from t0 to t2. Red symbols represent treatment changes during follow-up: crosses indicate a switch to enzalutamide, circles a switch to abiraterone plus prednisone, squares a switch to lutetium-177 therapy, and triangles a switch to chemotherapy. Differences between time points were analyzed using the Kruskal–Wallis test, with significant *p* values or trends (*p* = 0.1) indicated in the graphs. Pairwise differences between time points are marked (* *p* < 0.05).

**Figure 4 ijms-26-04669-f004:**
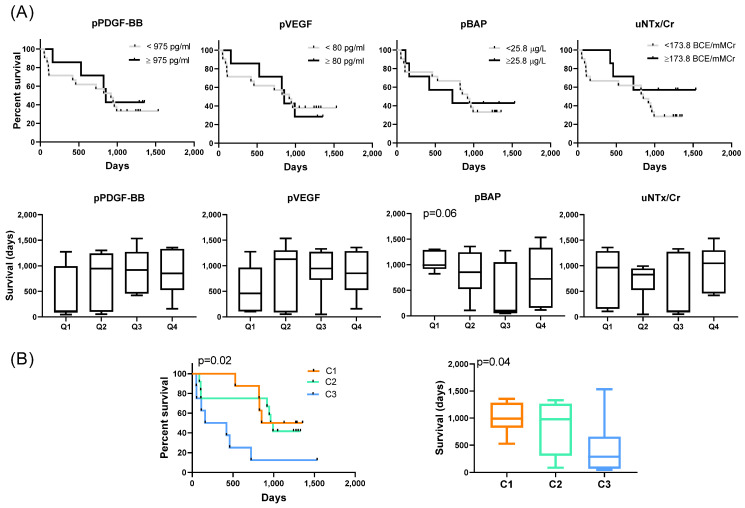
Survival analysis of mCRPC patients (*n* = 28) segregated into groups C1, C2, and C3. (**A**) Patients were categorized based on the Q3 value of the four molecules used by cluster classification: pPDGF-BB, pVEGF, pBAP, and uNTx/Cr (<Q3 and ≥Q3). No significant differences were observed in the survival curves or mean survival times across the quartiles (Q1, Q2, Q3, and Q4), except for pBAP, which showed a trend toward higher survival in Q1. (**B**) Comparison of survival curves for patients segregated as C1, C2, and C3 along with the survival times for each group. Survival curves were compared using log-rank test, and mean survival times were analyzed using ANOVA.

**Table 1 ijms-26-04669-t001:** Demographic and clinical characteristics of mCRPC patients at baseline, segregated into three groups (C1, C2, and C3).

Cluster	C1	C2	C3	*p*
Number of patients	8	12	8	
Age at Dx, years (mean ± SD)	68.63 ± 5.42	72.75 ± 8.47	74.88 ± 8.14	ns
Gleason score at Dx: <8 n (%)	2 (25)	5 (41.66)	1 (12.5)	ns
≥8 n (%)	6 (75)	6 (50)	4 (50)	ns
Metastasis at t0 *n* (%)				
Bone	6 (75.00)	10 (83.33)	7 (87.50)	ns
Visceral	2 (25.00)	3 (25.00)	4 (50.00)	ns
Adenopathic	4 (50.00)	9 (75.00)	2 (25.00)	ns
NLR (mean ± SD)	3.48 ± 1.54	5.81 ± 8.61	3.27 ± 1.34	ns
PLTs × 10^9^/L (mean ± SD)	248 ± 53.10	199 ± 74.01	218 ± 36.61	ns
Previous local therapy for prostate cancer at t0 *n* (%)	6 (75.00)	8 (66.66)	4 (50.00)	ns
Years from Dx at t0 (mean ± SD)	11.25 ± 6.92	8.00 ± 6.53	8.87 ± 6.89	ns
Treated with Abi at t0 *n* (%)	5 (62.5)	7 (58.33)	3 (37.50)	ns
Treated with BPs at t0 *n* (%)	0 (0.00)	2 (16.66)	1 (12.50)	ns
Years from Dx at exitus (median (range))	16 (14–20)	8 (1–19)	8.42 (2–20)	ns

Abbreviations: Dx: diagnosis; NLR: neutrophil-to-lymphocyte ratio; PLTs: platelet count; Abi: abiraterone + prednisone; BPs: bisphosphonates; ns: not significant; a total of 28 patients were included in the cluster analysis, as plasma samples were unavailable for one patient. As of October 2024, some patients in each group remain alive (C1: *n* = 4; C2: *n* = 6; C3: *n* = 1). Numeric variables were compared across the three groups using the Kruskal–Wallis test, while binary variables were analyzed using the Chi-square test.

**Table 2 ijms-26-04669-t002:** Comparison of levels of biomarkers in peripheral blood and urine based on the presence or absence of treatment or clinical characteristics.

		Abi	BPs	Sx	RT	Chemo	Gleason ≥ 8	M1 *n*
*n*	Yes No	17 12	4 25	11 18	12 17	7 22	17 8	>4 15 <4 9 No 5
# pPDGF-BB (pg/mL)	Yes No	134.5 ± 660.5 522.5 ± 436.7	442.3 ± 652.7 667.8 ± 575.9	669.2 ± 604.5 607.7 ± 572.2	620.5 ± 650.8 661.0 ± 534.1	524.5 ± 506.7 676.1 ± 599.9	634.4 ± 530.9 668.6 ± 766.4	487.4 ± 404.1 896.9 ± 737.8 625.2 ± 622.6
# pIGFBP-1 (pg/mL)	Yes No	6948 ± 5760 ^&^ **11,294 ± 7418**	2786 ± 1658 * **9534 ± 6785**	8246 ± 5827 9176 ± 7446	8209 ± 4291 9262 ± 8255	6453 ± 4576 9454 ± 7189	9853 ± 7625 7711 ± 5409	8595 ± 5878 6929 ± 4653 12,800 ± 11,158
# pBMP-2 (pg/mL)	Yes No	18.65 ± 29.89 71.04 ± 183.8	26.69 ± 26.05 42.84 ± 129.4	72.69 ± 190.8 20.67 ± 36.87	13.16 ± 16.72 62.06 ± 160.1	16.83 ± 20.59 47.73 ± 137.6	20.09 ± 36.71 99.67 ± 221.9	17.86 ± 32.88 83.87 ± 211.3 29.21 ± 44.24
# pVEGF (pg/mL)	Yes No	**30.98 ± 35.4** ^&^ 14.01 ± 28.69	25.62 ± 41.09 23.48 ± 33.19	32.92 ± 43.36 17.75 ± 24.35	16.80 ± 26.35 28.89 ± 37.58	28.26 ± 40.37 22.47 ± 32.01	29.40 ± 38.49 14.34 ± 18.95	10.45 ± 19.20 ^&^ **38.69 ± 38.89** 33.87 ± 44.54
pPSA (µg/L)	Yes No	287 ± 877 93.9 ± 195	202 ± 268 207 ± 729	48.3 ± 66.05 304 ± 857	324 ± 1045 124 ± 206	118 ± 216 235 ± 776	294 ± 875 23.7 ± 45.0	368.4 ± 932.5 37.62 ± 43.99 28.37 ± 38.65
pNSE (µg/L)	Yes No	12.7 ± 14.8 16.0 ± 19.6	**41.3 ± 36.0** ^&^ 9.77 ± 4.73	**21.6 ± 25.5** ^&^ 9.52 ± 4.49	13.7 ± 17.8 14.3 ± 16.5	**27.2 ± 30.9** ^&^ 9.95 ± 4.94	9.79 ± 3.26 19.3 ± 23.8	13.75 ± 17.92 17.63 ± 19.68 9.01 ± 2.89
pCgA (µg/L)	Yes No	167 ± 231 ^&^ **490 ± 691**	297 ± 475 302 ± 507	236 ± 336 340 ± 576	386 ± 678 241 ± 319	370 ± 506 279 ± 500	263 ± 528 378 ± 484	393 ± 624.2 250.8 ± 363.6 117.6 ± 64.47
pBAP (µg/L)	Yes No	27.2 ± 23.0 21.4 ± 15.3	24.5 ± 17.4 21.5 ± 13.7	18.7 ± 12.2 ^&^ **28.5 ± 23.2**	21.2 ± 12.7 27.4 ± 24.0	25.0 ± 18.2 24.7 ± 21.0	24.7 ± 22.8 20.0 ± 14.6	**24.88 ± 12.88 ^&^** 18.85 ± 13.40 13.5 ± 3.38
uPDGF-BB (pg/mL)	Yes No	22.6 ± 11.48 21.49 ± 7.79	17.94 ± 4.77 22.78 ± 0.02	19.34 ± 6.85 24.45 ± 11.89	18.20 ± 5.95 ^&^ **24.28 ± 11.6**	17.63 ± 3.20 23.64 ± 11.19	25.12 ± 11.66 20.13 ± 8.76	24.72 ± 10.99 19.91 ± 9.95 18.14 ± 4.84
uIGFBP-1 (pg/mL)	Yes No	1808 ± 4918 839.6 ± 1282	143.2 ± 44.9 * **844.0 ± 1534**	898.5 ± 1876 1894 ± 51.01	**3308 ± 6227** ^&^ 325.9 ± 366.9	1316 ± 2297 1531 ± 4557	748.1 ± 1634 2923 ± 6934	557.7 ± 1005 3292 ± 6583 287.0 ± 264.6
uBMP-2 (pg/mL)	Yes No	18.85 ± 4.73 15.71 ± 10.78	20.50 ± 5.22 17.34 ± 7.68	16.88 ± 7.96 18.37 ± 7.20	16.84 ± 6.66 18.3 ± 8.06	17.07 ± 6.58 17.92 ± 7.82	16.23 ± 6.28 18.23 ± 6.51	19.29 ± 7.26 14.23 ± 7.53 21.33 ± 5.24
uVEGF (pg/mL)	Yes No	90.92 ± 61.41 115.8 ± 147.6	119.7 ± 87.73 95.86 ± 100.8	87.08 ± 80.29 108.7 ± 110.5	101.1 ± 121.6 98.57 ± 83.73	101.9 ± 67.46 98.67 ± 108.2	**94.87 ± 76.43** ^&^ 56.61 ± 34.33	108.2 ± 115.5 90.0 ± 84.23 87.52 ± 53.15
uCr (mM)	Yes No	**10.5 ± 4.09** * 7.20 ± 3.57	6.37 ± 2.99 9.60 ± 4.20	8.15 ± 2.99 9.76 ± 4.72	8.31 ± 4.22 9.74 ± 4.15	7.37 ± 2.88 ^&^ **9.72 ± 4.40**	9.73 ± 4.20 8.61 ± 4.67	8.43 ± 4.92 ^&^ 10.22 ± 3.58 9.40 ± 2.58
uNTx/Cr (BCE/mM Cr)	Yes No	95.2 ± 72.8 201 ± 195	30.4 ± 11.6 ** **156 ± 148**	124 ± 125 148 ± 158	152 ± 154 130 ± 141	111 ± 164 147 ± 140	94.8 ± 69.3 163 ± 138	163.1 ± 167.3 126.0 ± 140.2 91.71 ± 59.39

Values are presented as mean ± SD of 29 patients. Data from only 28 patients were available for # variables. For each treatment group, Abiraterone (Abi), bisphosphonates (BPs), Surgery (Sx), radiotherapy (RT), and chemotherapy (Chemo), as well as bone metastases group (>4, <4, no), biomarker levels were compared. Pairwise comparisons were performed using the Mann–Whitney U test or Welch’s *t*-test. The Kruskal–Wallis test was used to compare the three groups in M1. Higher levels are in bold. If “yes” or “<4, or >4” presented higher levels than “no”, they are highlighted in dark orange (* *p* < 0.05) (in light orange if trending, & 0.05 > *p* < 0.1). If “yes” presented lower levels than “no”, it is highlighted in dark green (** *p* < 0.01), in medium green (* *p* < 0.05), and trends (&, 0.05 > *p* < 0.1) in light green.

## Data Availability

The data presented in this study are available on request from the corresponding author.

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
