# Peer review of "Pilot Investigation on Markers of Bone Metabolism, Angiogenesis, and Neuroendocrine Activity as Potential Predictors of Survival of Metastatic Prostate Cancer Patients with Bone Metastases"

_ijms, 2025, doi:10.3390/ijms26104669_

Round 1

Reviewer 1 Report

Comments and Suggestions for Authors

Title: Clusters Based on Markers of Bone Metabolism, Angiogenesis, 2 and Neuroendocrine Activity Predict the Survival of Prostate 3 Cancer Patients with Bone Metastases

Authors: M. Angels Ortiz, Georgia Anguera, Elisabet Cantó, Jose Alejandre, Josefina Mora, Ruben Osuna, María Mulet, Pradip Mora, Assumpta Antonijuan, Sofia Sánchez, Ona Ramírez, Vanessa Orantes, Pablo Maroto, Silvia  Vidal

In this scholarly article, the authors examined particular biomarkers in select prostate cancer patients exhibiting clusters of bone metastases and proposed that these biomarkers could assist in predicting survival outcomes. The manuscript is articulately composed, and the statistical analyses were appropriately executed.

Concerns:

  1. Why were PDGF-BB, IGFBP-1, BMP-2, VEGF specifically selected? While these markers are related to bone metabolism and angiogenesis, the rationale for excluding other known bone metastasis-related markers (e.g., PINP, OPG, RANKL, OPN) is unclear.
  2. In Table 1, the authors listed bone metastases as follows: 2 in C1, 3 in C2, and 3 in the C3 cluster. Earlier, they reported that 24 out of the 29 patients had bone metastases. Can you explain these discrepancies?
  3. In what manner might the limited number of patients in certain clusters (particularly in longitudinal follow-up for C3) influence the integrity of biomarker analyses? Could this potentially compromise the inferences made regarding biomarker dynamics??
  4. The authors may need to expand on how the type and timing of treatments (e.g., radiotherapy, bisphosphonates) impact longitudinal biomarker trends? Did patients switch or modify treatments between time points, potentially confounding biomarker levels?
  5. How confident can the authors be about the generalizability and reproducibility of the proposed clusters when only 29 patients were in this study? Could a larger validation cohort help substantiate their claims?
  6. The authors state bisphosphonate treatment reduces uNTx/Cr levels but also indicates that C3 has high uNTx/Cr despite bisphosphonate treatment presence. How do authors explain this apparent contradiction?
  7. Cluster descriptions sometimes are inconsistent. For example, Cluster 3 is described as having both elevated and lowered levels of biomarkers depending on the context (abstract vs. results section). This should be clarified clearly and consistently throughout the manuscript.

Author Response

  1. Why were PDGF-BB, IGFBP-1, BMP-2, VEGF specifically selected? While these markers are related to bone metabolism and angiogenesis, the rationale for excluding other known bone metastasis-related markers (e.g., PINP, OPG, RANKL, OPN) is unclear.

In our study, we selected as potential biomarkers molecules with strong and complementary roles in bone metabolism and angiogenesis, two fundamental processes in the development of bone metastases in metastatic castration-resistant prostate cancer (mCRPC): VEGF and PDGF-BB are key pro-angiogenic factors crucial for the establishment and maintenance of the bone metastatic niche. BMP2 is a central regulator of osteoblast differentiation and bone formation. IGFBP-1 influences bone metabolism and remodeling, which are frequently dysregulated in mCRPC. A sentence was added to explain this selection (lines 75-76). We are also aware of the clinical significance of additional markers frequently elevated in mCRPC: Elevated levels of bone resorption markers (N-telopeptide) and bone formation markers (bone alkaline phosphatase) have been consistently associated with poor survival outcomes. Markers like chromogranin A and neuron-specific enolase are indicative of neuroendocrine differentiation, which defines an aggressive disease phenotype and is linked to unfavorable prognosis.

Regarding the exclusion of other bone-related markers such as PINP, OPG, RANKL, and OPN, we agree that these are valuable markers of bone turnover and metastasis. However, broadening the panel to include these markers would have required an extended framework beyond the defined scope of this study. We plan to include a broader range of markers in future studies aimed at deeper phenotypic and prognostic characterization of mCRPC. We now clarify this limitation in the revised manuscript (lines 793-804).

  1. In Table 1, the authors listed bone metastases as follows: 2 in C1, 3 in C2, and 3 in the C3 cluster. Earlier, they reported that 24 out of the 29 patients had bone metastases. Can you explain these discrepancies?

In the first paragraph of the Results section, we reported that 24 out of the 29 patients included in our study had bone metastases at the time of sample collection (considered baseline or t0). In contrast, the data shown in Table 1 refer to the presence of bone metastases at the time of diagnosis. Specifically, when we report 2 patients with bone metastases in cluster C1, 3 in cluster C2, and 3 in cluster C3, we are referring to bone metastases present at initial diagnosis. To avoid confusion, we have modified Table 1 by replacing the data referring to the number of patients with bone metastases at diagnosis with the data corresponding to the number of patients with bone metastases at baseline (t0 = initiation of study and sample collection). We have also modified accordingly lines 226-229.

It is important to note that the study started at the time of sample collection and, in some cases, several years elapsed between diagnosis and sample collection. During this period, some patients who did not present with bone metastases at diagnosis developed them later. This explains the difference between the numbers presented in Table 1 and the overall number of patients with bone metastases described earlier in the Results section.

  1. In what manner might the limited number of patients in certain clusters (particularly in longitudinal follow-up for C3) influence the integrity of biomarker analyses? Could this potentially compromise the inferences made regarding biomarker dynamics??

We agree with the reviewer that longitudinal studies are particularly sensitive to the number of patients in the available time points. With only a few patients in C3 having follow-up measurements (e.g., 2 in t1 and t2), this compromises the ability to assess trends in biomarker dynamics over time in this group. We have noted this limitation in the results (lines 384-386). Future studies should aim to increase sample size and include additional biomarkers or longer follow-up periods to enhance the reliability and generalizability of the findings, as indicated in limitations (lines 793-804).

  1. The authors may need to expand on how the type and timing of treatments (e.g., radiotherapy, bisphosphonates) impact longitudinal biomarker trends? Did patients switch or modify treatments between time points, potentially confounding biomarker levels?

We appreciate the reviewer’s suggestion regarding the impact of treatment changes on longitudinal biomarker trends. In our analysis (Figure 3), we compared patients who changed treatment to enzalutamide, switched to other treatments, or remained on their initial treatment. However, we did not observe significant differences in biomarker trajectories between these groups.  

To explore the potential effect of treatment timing, we added the moment of treatment change to Figure 3. Despite this adjustment, no significant differences were found in biomarker trends across treatment groups. Biomarker changes were consistent within each cluster, but distinct across clusters, indicating that underlying cluster characteristics drove the biomarker dynamics rather than these particular treatment changes. This has been clarified in the Results section (lines 394-399).

Regarding uNTX/Cr levels, we observed differences between C1 and C2 (Figure 3), regardless of treatment changes. When analyzing the impact of previous treatments in the expression of each biomarker (Table 2), we found that bisphosphonates appeared to reduce uNTX/Cr levels. However, since no patients introduced or withdrew bisphosphonates during follow-up, we could not fully assess its impact.

We have added this point in the Discussion section (lines 793-804) and suggest that future research should investigate how the time elapsed after treatment change influences biomarker dynamics more comprehensively. Collecting samples prior to treatment changes would allow for a more comprehensive analysis, helping to better capture baseline biomarker levels and eliminate potential confounding factors associated with the timing and type of treatment changes.

  1. How confident can the authors be about the generalizability and reproducibility of the proposed clusters when only 29 patients were in this study? Could a larger validation cohort help substantiate their claims?

Our study is intended as a preliminary exploration of biomarker profiles and clustering patterns in a cohort of patients. As such, the findings represent an initial effort to identify potential subgroups of patients based on molecular profiles. We acknowledge that a limited sample size may impact the generalizability and reproducibility of the proposed clusters. However, we ensured that our data underwent systematic preprocessing and quality control prior to analysis. All biomarkers were measured using validated assays, and outliers or missing values were appropriately addressed. To classify patients based on their biomarker profiles, we employed a two-step cluster analysis using SPSS, a robust unsupervised algorithm that handles continuous variables and automatically selects the optimal number of clusters.

While the limited sample size and unequal distribution of patients across clusters may reduce the external robustness of the classification, the clustering process was data-driven, reproducible, and based on biologically relevant markers. A larger validation cohort would provide stronger evidence to substantiate our findings and enhance the external validity of the clusters. Additionally, collecting samples prior to treatment changes would improve the analysis by better capturing baseline biomarker levels and minimizing potential confounding factors related to the timing and type of treatment changes. This limitation is explained in detail in lines 793-804.

  1. The authors state bisphosphonate treatment reduces uNTx/Cr levels but also indicates that C3 has high uNTx/Cr despite bisphosphonate treatment presence. How do authors explain this apparent contradiction?

As indicated in the revised Table 1, only a small number of patients were receiving bisphosphonate treatment at baseline: 1 patient in cluster C3, 2 in cluster C2, and none in cluster C1. No patients initiated or discontinued bisphosphonate therapy during the study period. Given the very limited number of treated individuals and the absence of treatment changes over time, we are unable to assess any impact of bisphosphonates on uNTx/Cr levels in our cohort, other than the previous treatment. Therefore, the elevated uNTx/Cr observed in cluster C3 cannot be attributed to bisphosphonate treatment, and we do not consider this to represent a contradiction, but rather a reflection of baseline biological differences within the cluster.

  1. Cluster descriptions sometimes are inconsistent. For example, Cluster 3 is described as having both elevated and lowered levels of biomarkers depending on the context (abstract vs. results section). This should be clarified clearly and consistently throughout the manuscript.

Thank you for your observation. Despite abstract is correct, you are right in noting that the sentence stating “C3 consistently showed lower levels of these molecules, distinguishing it from the other two clusters” is inconsistent. This sentence should, in fact, refer to cluster C2, not C3. The characteristics of C3 are correctly described in the preceding sentences and are consistent with the description provided in the abstract. We will revise the manuscript accordingly to ensure clarity and consistency in the description of cluster-specific profiles. We will replace “C3” with “C2” in the relevant sentence of the manuscript (line 146).

Reviewer 2 Report

Comments and Suggestions for Authors

In attachment.

Author Response

Authors analysed a number of biomolecules (14) in 29 patients with prostate cancer and bone metastasis and defined 3 clusters of participants based on the results.

1- Patient group contained quite distinct cases in respect to interventions, therapy and time passed after therapy, and some of the laboratory results were significantly influenced by these factors, as authors discovered/stated themselves. It is hard (even impossible) to talk about clustering with so small number of cases defined by so many variables. Many more patients should be included in order to cluster (hundreds or even thousands).

We appreciate the reviewer’s thoughtful comment. Although we initially included 14 variables in our dataset, we used a two-step clustering algorithm, which incorporates an automatic feature selection step. As part of its internal optimization, the algorithm identified and retained only 4 variables as significantly contributing to the clustering solution. In a second analysis, the remaining variables were excluded due to low relevance or redundancy in defining distinct groups.

Two-step clustering is designed to handle mixed-type data and small-to-moderate sample sizes, and it employs statistical criteria to both select variables and determine the optimal number of clusters. This approach helps prevent overfitting and allows for a more robust, data-driven classification, even with a limited number of cases.

Nevertheless, we fully agree with the reviewer that these findings are exploratory and hypothesis-generating. We have revised the manuscript to:

  • Clarify the use and rationale of two-step clustering and emphasize the algorithm's variable selection mechanism. This is included in lines 875-883.
  • Include a more cautious interpretation of the clustering results and explicitly state the need for future validation in larger patient cohorts. This is now included in lines 779-781 and 793-804.
  1. Authors did not state what was the initial number of patients that were observed at the begining. There must have been many more befor making decision that 29 developed bone metastatis and should be further followed-up. In the section 2.1., it was stated that „At the time of diagnosis, 12 patients (41.38%) presented with metastatic disease.“ So, ...? What were inclusion and exclusion criteria? Later in the text it is said that at the time of study 24 patients had bone metàstasis - when the remaining 5 were diagnosed? The entire set-up is not clear.

We appreciate the opportunity to clarify these points. As requested, we have revised the manuscript to ensure the study setup is clearer. Regarding the initial number of patients, we agree that it is important to specify the total number of patients observed at the start. We have clarified that the inclusion period for the study started at baseline (t0), rather than at diagnosis, as noted in the revised manuscript. All consecutive patients who met the inclusion criteria were enrolled at baseline. The diagnosis was recorded retrospectively to assess the time since diagnosis and prior treatments. This point has been clarified in lines 819-821.

To further clarify, the 29 patients were included and at baseline, 24 out of these 29 patients had bone metastasis, as detailed in the revised Table 1.

  1. Clusters were defined for each biomarker individually. Clustering is the method to stratify population taking into account combined markers – can authors cluster patients using combined biomarkers which profiled as potentialy important in prediction?

We appreciate the reviewer’s comment. We would like to clarify that the patient clustering (C1, C2, C3) was based on a combination of biomarkers that emerged as potentially predictive. This unsupervised clustering approach enabled stratification of patients based on integrated biomarker profiles. We subsequently used these clusters in survival analyses, which revealed differences in clinical outcomes. Levels of individual biomarkers were associated with clinical characteristics and prior treatments to understand the evolution of these levels. We have now clarified this approach in the revised manuscript. Information will be found in lines 90-93 and 98-100.

  1. If the idea was to define and compare clusters, why no-clustering approach was given in Table 2?

We agree that including a clustering approach in Table 2 could be of interest. However, as we now clarify in a new paragraph added at lines 243-246 and a new Table S1, a formal statistical comparison between clusters was not performed due to the limited number of patients in some groups, which would compromise the robustness and reliability of the results.

  1. Axis in diagrams should have units (where missing) and no data should be reported as O.D. – the concentration of each marker is in some units.

Thank you for your valuable feedback. We have revised Figures and Figure Supl.1 to include the corresponding concentration units. Additionally, we have updated the values in Table 2, replacing OD by units to ensure clarity and consistency. We appreciate your careful review and suggestions to improve the manuscript.

  1. Figure 2B has no explanation what the size of circles represents.

We appreciate your comment. As suggested, we have now included a descriptive sentence in the legend of Figure 2 to clarify the graphical representation of the correlation coefficients.

  1. In Table 1, only 28 patients are presented.

Thank you for your valuable comment. In our study, we included 29 patients. However, for one of these patients, we were unable to determine the plasma levels of PDGF-BB, IGFBP-1, BMP-2, and VEGF at t0. As a result, when we introduced the biomarker data into the statistical software for clustering, the system automatically excluded this patient from the analysis, and they were not assigned to any cluster. However, the other variables of this patient were included in the rest of the manuscript.

To clarify this point in the manuscript, we have modified lines 128-129, adding a paragraph explaining what happened with this patient and his characteristics. We believe this change will enhance the clarity of our study. Additionally, we have included the number of patients in the legends of Figures 1, 2, and 3 as well as in the footnotes of Tables 1 and 2, to ensure clarity and transparency regarding the availability of data.

  1. Authors stated that „... patients in C3 tended to have a higher prevalence of bone metastases at the baseline than those in C1 and C2.“ – it is not relevant to say that by comparing 3 to 2 cases.

Following a helpful comment from another reviewer, we have updated Table 1 to report the number of metastatic sites at the time of sample collection (t0) rather than at diagnosis, as we believe this timepoint is more relevant for the interpretation of our biomarker data. Consequently, we have removed the sentence at line 177 stating that "patients in C3 tended to have a higher prevalence of bone metastases at baseline than those in C1 and C2," as this comparison was based on very small numbers and is no longer aligned with the revised data. We now state that there were no significant differences in the frequency of metastases between clusters at t0 and we mentioned at line 227-229.

  1. In Figure 3, the legend informes the reader what type of therapy change happened with individual patients – another variable not mentioned previously in the text; additionally disabling any firm conclusion with 29 patients.

We thank the reviewer for this insightful comment. We have now revised Figure 3 to explicitly indicate the timing and nature of treatment changes for individual patients during the follow-up period. While the sample size limits the possibility of performing stratified statistical analyses, descriptive evaluation showed similar biomarker trajectories in patients who remained on the same therapy and those who switched. Importantly, we observed that the dynamics of key biomarkers—particularly uNTx/Cr—remained distinct across clusters C1 and C2, independently of whether patients changed treatment and of the timing of such changes (lines 479-485).

These observations support the idea that the biological characteristics defining each cluster—rather than these treatment variations. Nevertheless, we acknowledge the heterogeneity in treatment as a potential confounder and have emphasized this limitation more clearly in the revised Results (lines 466-468) and Discussion sections (lines 793-804).

  1. A large part of Discussion is not necessary – there is no need to revise the physiological role of individual markers since the topic of the article is clustering of the already established markers.

We have reduced the discussion accordingly.

  1. Ethical approval number is missing in section 4.1

We appreciate the reviewer’s comment. The ethical approval number (IIBSP-P120-2018-20) was inadvertently omitted. We have now included it in section 4.1 as required.

Reviewer 3 Report

Comments and Suggestions for Authors

The study is interesting and presents a valuable investigation into the use of plasma and urinary biomarkers for patient stratification and prognostic evaluation, which is highly relevant.

However, several aspects should be improved and clarified.

It is important to discuss the experimental sample size and its limitations. Some clusters, such as C1 and C3, include only 8 patients, which leads to drastic changes in survival curves, as shown in Figure 4.

The silhouette score values for different numbers of clusters are not presented.
Was the score for K=3 substantially better than for K=2 or K=4? Without this data, the selection of three clusters appears arbitrary. A graphical representation of the clusters could be helpful for visualization.

Did the authors test other validation methods for clustering?

Additionally, the introduction does not mention treatment approaches for castration-resistant prostate cancer (CRPC) with bone metastases, particularly the use of radium-223. It is noteworthy that none of the patients received this therapy, even though it is indicated for bone-only metastatic prostate cancer. According to Table 1, clusters C1 and C2 included only patients with bone metastases.

In the methods section, the baseline time point (t0) is defined as the sample collection time for patients under androgen deprivation therapy (ADT), but the duration of ADT prior to sample collection is not specified. This information is crucial, as ADT duration could influence the levels of the biomarkers analyzed. the same for Dx, therapy for prostate cancer, Abi, BPs.

Similarly, the metastatic burden is not reported, which is highly relevant because the extent of metastasis could directly impact biomarker levels.

Additionally, some biomarkers are reported in optical density (OD) units. This makes it difficult to determine whether they fall within the detection limits of the ELISA technique. It would be preferable to report these values as absolute concentrations (Figure 3).

Finally, it is important to clarify whether abiraterone treatment was co-administered with corticosteroids. The manuscript describes renal effects associated with abiraterone, which are generally prevented by co-administration with prednisone.

Author Response

  1. It is important to discuss the experimental sample size and its limitations. Some clusters, such as C1 and C3, include only 8 patients, which leads to drastic changes in survival curves, as shown in Figure 4.

We thank the reviewer for this important observation. Indeed, the sample sizes for clusters C1 and C3 (n=8 each) are limited, which may affect the stability of survival curve estimates and increase the impact of outliers. We acknowledge this as a limitation and have now added a discussion point to reflect the exploratory nature of the clustering analysis (lines 793-804). Despite the small size, clustering was performed following rigorous steps, and we have updated all quality control measures, including PCA Distribution (now in Figure 1), silhouette score assessment, and the strategy used to select the most significant variables (explained in Material and Methods, lines 964-971). In addition, it is important to highlight that the observed differences in survival were statistically significant and consistent with the distinct biomarker profiles of the clusters, supporting their potential biological relevance. We also emphasize that these findings require validation in an independent and larger cohort to confirm their prognostic value (in limitations of Discussion, lines 793-804).

  1. The silhouette score values for different numbers of clusters are not presented.
    Was the score for K=3 substantially better than for K=2 or K=4? Without this data, the selection of three clusters appears arbitrary. A graphical representation of the clusters could be helpful for visualization.

We thank the reviewer for this important comment regarding the cluster selection criteria. In the initial two-Step Cluster Analysis using all 14 variables across 28 patients, the model returned a silhouette measure of cohesion and separation of 0.3, indicating poor to fair clustering quality. Importantly, this analysis also highlighted pPDGF, pVEGF, pBAP, and uNTx/Cr as the variables with the greatest predictor importance.

To improve cluster quality and biological interpretability, we repeated the clustering using only these four variables. This refined model showed a substantial improvement, with a silhouette score of 0.6, indicating good clustering quality. While both 3- and 4-cluster solutions achieved the same silhouette value (0.6), the 4-cluster solution included a very small group (n < 5), which we considered statistically unstable and clinically less informative. Based on these findings, we selected the 3-cluster solution for downstream analysis. We have added this explanation to the Methods section (lines 967-975) and included a PCA plot (Figure 1) to visually represent the clustering structure.

  1. Did the authors test other validation methods for clustering?

We appreciate the reviewer’s concern regarding additional clustering validation methods. Due to the exploratory nature and limited sample size of our study (n=28), we did not apply other validation methods, such as bootstrapping or external validation techniques. However, we believe that the use of silhouette values, alongside the observed clinical outcome correlations (such as survival differences), supports the biological and statistical relevance of the identified clusters.

We have now included this explanation in the Discussion section and plan to apply additional validation techniques in future studies with larger cohorts (lines 793-804).

  1. Additionally, the introduction does not mention treatment approaches for castration-resistant prostate cancer (CRPC) with bone metastases, particularly the use of radium-223. It is noteworthy that none of the patients received this therapy, even though it is indicated for bone-only metastatic prostate cancer. According to Table 1, clusters C1 and C2 included only patients with bone metastases.

Thank you for your comment. We have now mentioned treatment approaches for mCRPC patients (lines 53-55). In our hospital, patients included in the study were those treated with first-line ARPIs, which is the standard of care. Radium-223 is typically used in later treatment lines; therefore, none of the patients had received Radium-223 prior to their inclusion in the study (explained in Material and methods, lines 906-908).

Thank you for your comment in regard to the bone metastases. As a result of a suggestion from another reviewer, we have updated Table 1 to reflect the number and sites of metastases at the time of sample collection (t0), rather than at the time of diagnosis, as this information is more relevant for interpreting the biomarker data. Additionally, we would like to clarify that, at diagnosis, some patients did not present with bone metastases but rather with nodal involvement (lymph node metastases). The metastases information at diagnostic was originally reported in Table 1 and may have led to the misunderstanding. The revised table with metastases information at baseline now better reflects the clinical status of the patients when samples were collected.

  1. In the methods section, the baseline time point (t0) is defined as the sample collection time for patients under androgen deprivation therapy (ADT), but the duration of ADT prior to sample collection is not specified. This information is crucial, as ADT duration could influence the levels of the biomarkers analyzed. the same for Dx, therapy for prostate cancer, Abi, BPs.

The requested information regarding the duration of ADT prior to sample collection, as well as details on Dx, therapy for prostate cancer, Abi, and BPs, has been compiled and is now provided in Table S2. On the other hand, in Figure 2, we have acknowledged how the duration of the different treatments may influence the levels of the analyzed biomarkers. Although we did not previously indicate the treatment durations, we examined the correlations between treatment duration and biomarker levels, as shown in this figure. We have also updated the manuscript accordingly by adding a reference to this table in Section 4.1. We hope that this addition addresses your concern, and we sincerely appreciate your thorough and constructive review.

  1. Similarly, the metastatic burden is not reported, which is highly relevant because the extent of metastasis could directly impact biomarker levels.

We agree that metastatic burden is an important variable that can impact biomarker levels. However, we did not apply the CHAARTED criteria to define metastatic volume, as these criteria have not been validated in patients with mCRPC, which is the population included in our study. To address your concern, we have revised Table 2 to include biomarker levels stratified by metastatic burden at the time of sample collection. Specifically, we now differentiate between patients with > 4 bone metastases, those with < 4 bone metastases, and those without bone metastases at baseline (lines 455-457). We believe this provides a more relevant and accurate assessment of the potential impact of metastatic extent on biomarker expression in our cohort.

  1. Additionally, some biomarkers are reported in optical density (OD) units. This makes it difficult to determine whether they fall within the detection limits of the ELISA technique. It would be preferable to report these values as absolute concentrations (Figure 3).

Thank you for your valuable feedback. We have revised Figures to include the corresponding concentration units. Additionally, we have updated the values in Table 2, replacing OD by units to ensure clarity and consistency. We appreciate your careful review and suggestions to improve the manuscript.

  1. Finally, it is important to clarify whether abiraterone treatment was co-administered with corticosteroids. The manuscript describes renal effects associated with abiraterone, which are generally prevented by co-administration with prednisone.

All patients in our study who were treated with abiraterone received concomitant prednisone from the start of treatment. To clarify this, we will update the manuscript to specify "Abiraterone + prednisone" instead of just "Abiraterone."  We agree that the co-administration of prednisone with abiraterone generally helps prevent mineralocorticoid-related adverse effects (hypertension and hypokalemia related to increase of creatinine levels). However, as reported in the phase 2 study by Attard et al. published in JAMA Oncology [now reference 33], which evaluated the safety of different glucocorticoid regimens combined with abiraterone, Attard et al reported that treatment with abiraterone and prednisone was associated with increased creatinine levels in some patients. We add this information in discussion section (lines 611-613) to better contextualize our findings.

Reviewer 4 Report

Comments and Suggestions for Authors

This manuscript provides a thoughtful analysis of biomarker-driven clustering in prostate cancer with bone metastases. Your research delivers significant prognostic insights; however, certain aspects need clarification. In particular, the definition of Cluster C2 and the statistical robustness given the small sample size require attention. Below are detailed comments to help strengthen your manuscript.

  1. The authors have not clearly defined how they classified patients into Cluster C2 in result 2.2. The distinction between C1, C2, and C3 is clear for C1 and C3, but C2 lacks precise characterization.
  2. The authors have shown the correlation between different biomarkers: pPDGF-BB and pVEGF; pIGF-BP1 and pCga; pVEGF and uCr; pBAP and uNTx/Cr; and pNSE and pBAP. However, they have not described the relevance of these results in the results section.
  3. The terms "castration-resistant prostate cancer (mCRPC)" and "metastatic prostate cancer with bone involvement" are sometimes used interchangeably. Please clarify whether all patients in the study were mCRPC.
  4. The study includes a relatively small cohort (n = 29), which may impact the robustness of the clustering and statistical analyses. Were any power calculations conducted to justify the adequacy of the sample size? Given the limited number of patients at later time points (e.g., only two patients in C3 at 12 months), the authors should discuss the potential limitations of their conclusions.

Author Response

  1. The authors have not clearly defined how they classified patients into Cluster C2 in result 2.2. The distinction between C1, C2, and C3 is clear for C1 and C3, but C2 lacks precise characterization.

We thank the reviewer for this insightful comment. We have revised Section 2.2 to provide a clearer characterization of Cluster 2. Specifically, we corrected an error where C3 was mistakenly referenced instead of C2 (line 146). C2 is now properly described as representing a transitional profile, with intermediate biomarker levels that lie between the angiogenesis-dominant pattern of C1 and the bone turnover–dominant signature of C3 (lines 589-590).

  1. The authors have shown the correlation between different biomarkers: pPDGF-BB and pVEGF; pIGF-BP1 and pCga; pVEGF and uCr; pBAP and uNTx/Cr; and pNSE and pBAP. However, they have not described the relevance of these results in the results section.

We have subsequently described the relevance of the observed correlations and include a paragraph in Section 3 summarizing this information (lines 595-599) and the respective new references (20-24). The observed correlations suggest distinct pathogenic mechanisms, with PDGF-BB and VEGF linked to neovascularization and metastasis, pIGFBP-1 and pCgA associated with poor prognosis, and pNSE inversely related to bone metabolism markers, indicating a neuroendocrine tumor profile.

  1. The terms "castration-resistant prostate cancer (mCRPC)" and "metastatic prostate cancer with bone involvement" are sometimes used interchangeably. Please clarify whether all patients in the study were mCRPC.

We acknowledge the inconsistency in terminology and have revised the manuscript to ensure clarity. All patients included in the study have metastatic castration-resistant prostate cancer (mCRPC) with most of them with bone metastases. We have adjusted the wording throughout the text and the title of the manuscript to accurately reflect this distinction. We appreciate your careful review and valuable feedback.

  1. The study includes a relatively small cohort (n = 29), which may impact the robustness of the clustering and statistical analyses. Were any power calculations conducted to justify the adequacy of the sample size? Given the limited number of patients at later time points (e.g., only two patients in C3 at 12 months), the authors should discuss the potential limitations of their conclusions.

We appreciate the reviewer’s concern regarding the small sample size and its potential impact on the robustness of clustering. As this study was exploratory and data-driven, power calculations were not applicable in the classical sense. Instead, to ensure robustness and interpretability, we applied a two-step clustering method and tested multiple cluster solutions (2 to 4 groups). The selection of three clusters was based on a balance between clustering quality (which remained similar across solutions) and practical constraints, such as ensuring that each cluster included enough patients (more than five) to allow meaningful comparisons (lines 970-984). Although this approach does not rely solely on formal model-fit statistics, it integrates statistical validity with biological and clinical interpretability. In addition, we now include a Principal Component Analysis (PCA) plot (Figure 1) to visually support the distinct clustering pattern, which further confirms the relevance and separation of the identified groups.

Finally, we have added a paragraph in the Discussion section acknowledging the limited cohort size, particularly the reduced number of patients in some clusters at the 12-month time point, and how this may impact the strength and generalizability of our conclusions (lines 793-804).

Round 2

Reviewer 2 Report

Comments and Suggestions for Authors

In attachment

Author Response

  • Authors offered certain explanations for their approach and conclusions. However, none of these comments and changes in the manuscript overcome the initial problem – the number of cases is too small (28 or 29), with a number of variables that affect results. As I have previously stated, clustering is meaningfull with hundreds or even thousands of patients. Likewise, conclusions concerning results are not reliable with 29 cases. The only way to give the meaning to the results is to inlude much larger group of examinees. Nothing less.

We have fully acknowledged the limitation of our relatively small sample size in the discussion of the manuscript. It is true that clustering methods are ideally applied to larger datasets. However, recent systematic simulations by Dalmaijer et al. (2022) demonstrate that clustering can be statistically valid even with small sample sizes, provided that subgroups are reasonably well separated.

Our choice of the two-step clustering algorithm was deliberate, as it is specifically designed to accommodate small-to-moderate sample sizes and mixed data types. Despite the limited sample size, the algorithm identified three distinct clusters with statistically significant differences across all four variables used. This internal consistency suggests that the clustering structure is not random and may reflect underlying biological or clinical heterogeneity. Given the complexity of mCRPC and the push toward individualized patient care, we believe these preliminary subgroupings are clinically plausible and merit further exploration.

We emphasize that this is an exploratory, hypothesis-generating study—not intended to establish definitive clinical subgroups, but to identify patterns that could inform larger-scale investigations. While we agree that validation in a much larger cohort is necessary before clinical implementation, we also recognize that generating such data entails significant logistical and financial challenges. In this context, smaller-scale exploratory studies like ours remain essential to initiate new hypotheses and guide future research efforts.

We hope that this clarification underscores the rationale and value of our approach, despite the acknowledged limitations.

Dalmaijer ES, Nord CL, Astle DE. Statistical power for cluster analysis. BMC Bioinformatics. 2022 May 31;23(1):205.

  • My comment that there must have been many more patients before making decision which to include was not answered properly. There is no need for clarification for those included, but statement how many patients presented in that hospital during the collection time (which was...?), that had prostate cancer with metastasis, and how those included in the study were chosen. Prostate cancer is a relatively frequent disease, Barcelona is a big city and 29 cases may be diagnosed probaly in several weeks. What puzzels me further is the ethical approval number - IBSP-P120-2018-20? Does it date back from 2018? If so, the number of appropriate patients, even if the study started later on, must be much larger.

We thank the reviewer for raising this important point regarding patient selection and recruitment. In the revised manuscript, we have now clearly stated that the inclusion period was from June 2020 to February 2022. During this time, all consecutive patients diagnosed with metastatic castration-resistant prostate cancer (mCRPC) and treated at Hospital de la Santa Creu i Sant Pau were assessed for inclusion.

Patients were selected based on predefined inclusion and exclusion criteria, detailed now in the Methods section: 

Inclusion criteria: confirmed mCRPC diagnosis, ECOG performance status ≤1, estimated life expectancy >12 weeks, and no major laboratory abnormalities.

Exclusion criteria: presence of brain metastases and participation in or eligibility for clinical trials.

Now, we have included a paragraph in Methods section, indicating these criteria (lines 361-366)

As the reviewer emphasized, our hospital is a tertiary referral center in a large metropolitan area (Barcelona), with a significant population but also where approximately 50% of prostate cancer patients are referred for or enrolled in clinical trials. Consequently, many potential candidates were not eligible for inclusion due to their participation in investigational protocols.

In addition, the inclusion period overlapped with the COVID-19 pandemic, which significantly disrupted routine clinical activity, reduced patient visits, and limited access to in-person study procedures. This global health crisis inevitably impacted recruitment and contributed to the reduced size of the cohort.

Regarding the ethical approval number (IBSP-P120-2018-20), we clarify that the protocol was approved in 2018 as part of a broader translational research initiative. However, recruitment for the present study began in June 2020, in line with internal project timelines and logistic implementation.

We trust this explanation clarifies how patients were selected and why the final sample size, though limited, is justified within the specific context of our study.

Reviewer 3 Report

Comments and Suggestions for Authors

I thank the authors for the clarity of their responses and the improvements incorporated into the manuscript. I believe the revised version now includes substantial enhancements that support its approval without further revisions, as the conclusions are well aligned with the results and appropriate mention is made of the study’s limitations.

Author Response

Thank you!